# Dynamics of Sediments in Reservoir Inflows: A Case Study of the Skalka and Nechranice Reservoirs, Czech Republic

**Jan Pacina** [1,*] **, Zuzana Lenďáková** [2] **, Jiří Štojdl** [1] **, Tomáš Matys Grygar** [1] **and Martin Dolejš** [3]

1   Department of Geoinformatics, Faculty of Environment, J. E. Purkyně University,
40092 Ústí nad Labem, Czech Republic; jiri.stojdl@ujep.cz (J.Š.); grygar@iic.cas.cz (T.M.G.)
2   Department of Geology, Faculty of Science, Palacký University Olomouc, 77146 Olomouc, Czech Republic;
zuzana.lendakova@upol.cz
3   Department of Geography, Faculty of Science, J. E. Purkyně University,
40092 Ústí nad Labem, Czech Republic; martin.dolejs@ujep.cz
*   Correspondence: jan.pacina@ujep.cz; Tel.: +420-608-134-253

**Abstract:** A wide variety of geographic information system tools and methods was used for pre-dam topography reconstruction and reservoir bottom surveying in two dam reservoirs in the Ohře River, Czech Republic. The pre-dam topography was reconstructed based on archival aerial imagery and old maps. The benefits and drawbacks of these methods were tested and explained with emphasis on the fact that not all processed archival data are suitable for pre-dam topography modeling. Bathymetric surveying of a reservoir bottom is presently routine, but in this study, we used a wide combination of bathymetric mapping methods (sonar, ground penetration radar, and sub-bottom profiler) and topographic survey tools (LiDAR and photogrammetry), bringing great benefits for bottom dynamic analysis and data cross-validation. The data that we gathered made it possible to evaluate the formation of the inflow deltas in the reservoirs studied and assess the sediment reworking during recent seasonal drawdowns. A typical inflow delta was formed in the deeper of the two studied reservoirs, while the summer 2019 drawdown caused the formation and incision of a temporary drawdown channel and erosive downstream transport of approximately 1/10 of the delta body thickness in approximately 1/10 of the delta transverse size. No inflow delta was formed in the shallower of the studied reservoirs, but unexpectedly extensive sediment reworking was observed in the inflow part of the reservoir. Both the studied reservoirs and the pre-dam river floodplain have accumulated historical contamination by risk elements such as As, Hg, Pb; thus, the enhanced erosion of existing sediment bodies expected in the future, owing to more frequent droughts and global climate change, will endanger the ecological quality of the water and solids outflowing from the reservoirs.

**Keywords:** dam reservoirs; reservoir management; sub-aquatic topography; DEM precision; pollution transport

---

## 1. Introduction

The current discussion on river damming versus freely flowing rivers is mostly focused on the continuity of sediment and nutrient transfers through the fluvial systems and the sustainability of fluvial landscapes and biomes [1]. An inevitable problem is reservoir filling with sediments, which considerably shortens the effective performance of the reservoir [1–4]. To limit this problem, flushing (sluicing) operations are performed [1,2,5]. Nevertheless, reservoirs with severe historical pollution [6–8] can experience rather opposite concerns—their sustainable management should include the prevention

of sediment loss to protect downstream aquatic systems. Knowledge of sediment deposition and erosion (reworking) patterns is required for both of these cases. Therefore, improved insights into the morphodynamic mechanisms are needed in reservoir research [1].

Sediment deposition in reservoirs has been studied more intensively than sediment erosion or reworking. The spatial pattern of sediment deposition is difficult to predict, with huge impacts of hydrological events [4,9] and different depositional patterns at shorter and longer timescales [9]. Prograding deltas contain most the sediments transported by tributary streams [1,4,10]. Under low flow conditions, the most common process is deposition in stepwise prograding deltas along with the incision of submerged channels, which results in a long-term increase in near-dam deposition [4,9,11]. Erosion, including slumps in the inflow delta front, channel, and gully erosion in delta tops, and subaerial and subaquatic bank instabilities, is typically more spatially heterogeneous than deposition [1]. Regarding historical pollution, the erosion of inflow deltas is of particular interest, because their sediment bodies are the most contaminated and morphologically the most dynamic sedimentary body in reservoirs [8].

The typical methodological design in morphodynamic studies on reservoir deposition employs a comparison between pre-dam topography and geophysical (sonar, ground-penetrating radar, laser scanning (LiDAR)) imaging of the actual bottom topography [1,3,4,8,10]. Advances in geographic information system (GIS) capabilities coupled with geophysical imaging have made it possible to conduct detailed studies of the spatiotemporal evolution of inflow deltas in reservoirs.

Having the pre-dam topography in the form of a digital elevation model (DEM) is crucial for analysis of the sedimentary patterns in reservoirs using GIS methods [12–14]. The former landscape can be reconstructed based on suitable historical maps, processed archival aerial photographs, or satellite images [15–20]. The need of DEM creation narrows the choice of archived spatial data to old maps containing elevation information or archived aerial imagery with sufficient overlap [14,16,21–23]. Several methods for noise removal and DEM smoothing suitable for archival data processing have been introduced [23,24].

The topography of the reservoir bottom can be studied using different data collection methods. One very common method is the use of the shipborne sonar sensors [25–29]. The density of surveyed data is dependent on the survey plan defined and the type of sensor (single-beam, multi-beam, or side-scan) used. Other remote sensing techniques, including satellite imagery, LiDAR, and photogrammetry, can be used to map dam topography under specific conditions [27,29–31]. Another option for visualizing bottom topography (bathymetry) is geophysical surveying. In this study, ground-penetrating radar (GPR) and a parametric sub-bottom profiler were used. Although GPR was not primarily used in water bodies, many studies have proven the value of GPR in such research. GPR can be used to determine the bathymetry and water volume, monitor sediment infilling, thickness, distribution throughout the water reservoir, and assess the volume of sediments or sediment characterization [4,10,32–35]. Sub-bottom profiling is used widely in sea-bed surveying [36–38] and has also been used successfully in freshwater environments [8,39,40].

Our work focused on the sediment dynamics of inflows to the Skalka and Nechranice reservoirs in the Ohře River. The reservoirs have been impacted by severe pollution by Hg from the former Marktredwitz Chemical Factory (MCF), [6,8,41,42] and several heavy metals and As from historical ore mining in the Krušné Hory/Erzgebirge Mountains along the Czech–German border [6,8,41]. The reservoirs have efficiently trapped the historical pollution, but this efficiency could decline due to sediment reworking during drawdowns [42] and stepwise downstream transport of sediments [42]. Our study aimed to answer the questions of whether the two studied inflows retain the trapped sediments permanently and to what extent the sediments are reworked. Our hypothesis was that the inflow deltas are not persistent sinks of sediments (and pollutants) but are quickly (seasonally) reworked. This hypothesis is in line with observation by [43]. We used the occasions of seasonal drawdowns in the reservoirs caused mainly by European summer droughts in 2018 and 2019. Great emphasis was given to the portfolio of spatial datasets. The quality of the datasets used for modeling the submerged

terrain was more difficult to determine than in photogrammetry, but more methods and repetitions of analyses could be used. Historical maps and archival aerial photographs were used to reconstruct the pre-inundation landscape. A combination of methods, including sonar bathymetry, sub-bottom profiler, GPR, LiDAR, and aerial imaging of recent surfaces made it possible to evaluate the collected data critically. Sufficient volumes of data and types of datasets were gathered to distinguish and omit data with inadequate quality. The results showed that some of the methods successfully applied for historical georelief modeling are not suitable for modeling the pre-dam topography in the study areas of this paper.

## 2. Methodology

### 2.1. Study Area

The Ohře River catchment and its pollution history (Figure 1) have been described in detail previously [6,41,42]. The Ohře River total length is 316 km, the Skalka Dam is at the river length of 242 km, and the Nechranice Dam at the river length of 103 km (river km is the distance from the river confluence with the Labe River). he Skalka Reservoir is severely polluted by Hg from a former chemical factory in Germany. The Nechranice Reservoir has received secondary pollution by As, Pb, and several other risk elements from mining, which peaked in the 16th century.

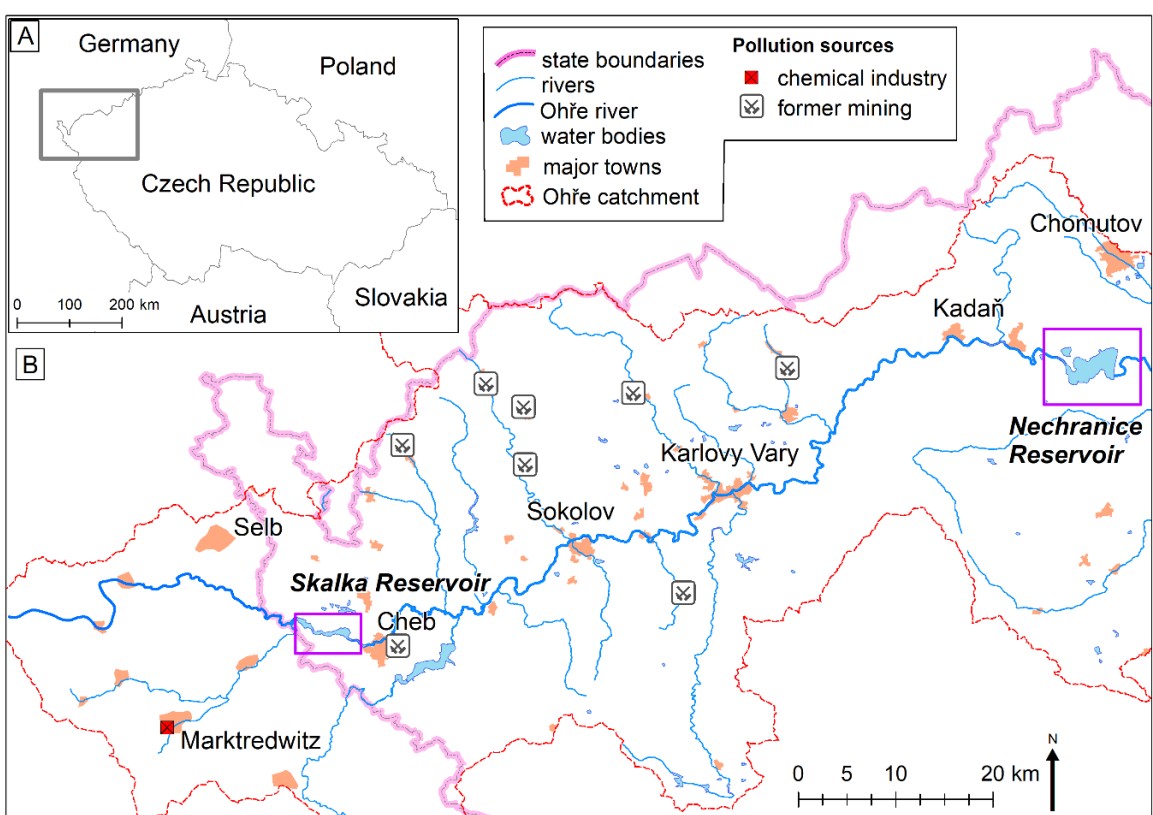

**Figure 1.** Map of the upper and middle reaches of the Ohře River with the positions of major historical sources of pollution by risk elements and the two studied reservoirs.

The Skalka Dam was constructed between 1962 and 1964. Under normal water level, its area is 3.4 km$^2$, its volume is 19.6 millions m$^3$, and the maximal depth of water near the dam is 12 m. The area shown in Figure 2 is subjected to varying water levels in seasonal drawdowns, and it was a subject area in this study. The reservoir had water levels of approximately 437.5 m a.s.l. between December to February and approximately 441.5 m a.s.l. between June and September between 2015 and 2019. Both the pre-dam floodplain and the reservoir have accumulated Hg from the MCF to the level that



the consumption of fish from the reservoir has been prohibited since the beginning of instrumental Hg analyses [42]. The Hg pollution is accompanied by Zn contamination, exceeding the background concentrations by several times [41]. Sediment sampling by gravity cores and analyses by laboratory X-ray fluorescence have been described in preceding papers [8,44]. The sediment is mostly sandy with silt admixture.

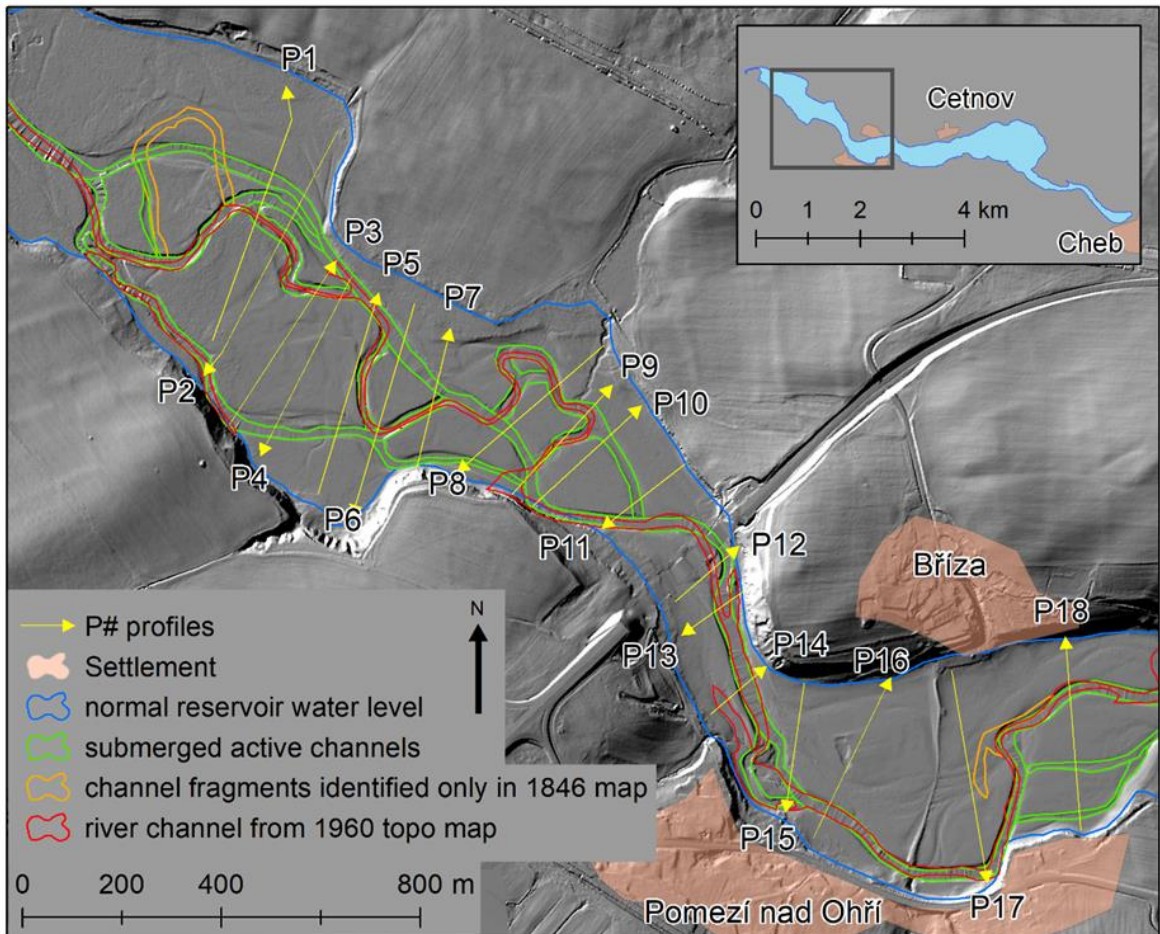

**Figure 2.** Detailed maps of the inflow to the Skalka Reservoir with positions of transect profiles (P#) and positions of banklines of historical and current channels of the Ohře River. The shaded light detection and ranging (LiDAR) data show the 2011 drawdown.

The Nechranice Dam was constructed between 1961 and 1968. Under normal water level, its area is 13.4 km$^2$, its volume is 287,600 m$^3$, and its maximal depth is 46 m. The reservoir had water levels from approximately 264 and 269 m a.s.l. between 2015 and 2019. The reservoir, particularly its inflow (Figure 3), accumulates risk element (As, Cu, Pb, Zn) pollution from the historical mining sites shown in Figure 1, as reported previously [6,8]. Our preceding study [8] revealed a considerable thickness of inflow delta and its importance for contamination storage, but the delta dynamics were not studied in detail there. The sediment contains mainly sand in the inflow and mainly very fine sand and silt in the basin.

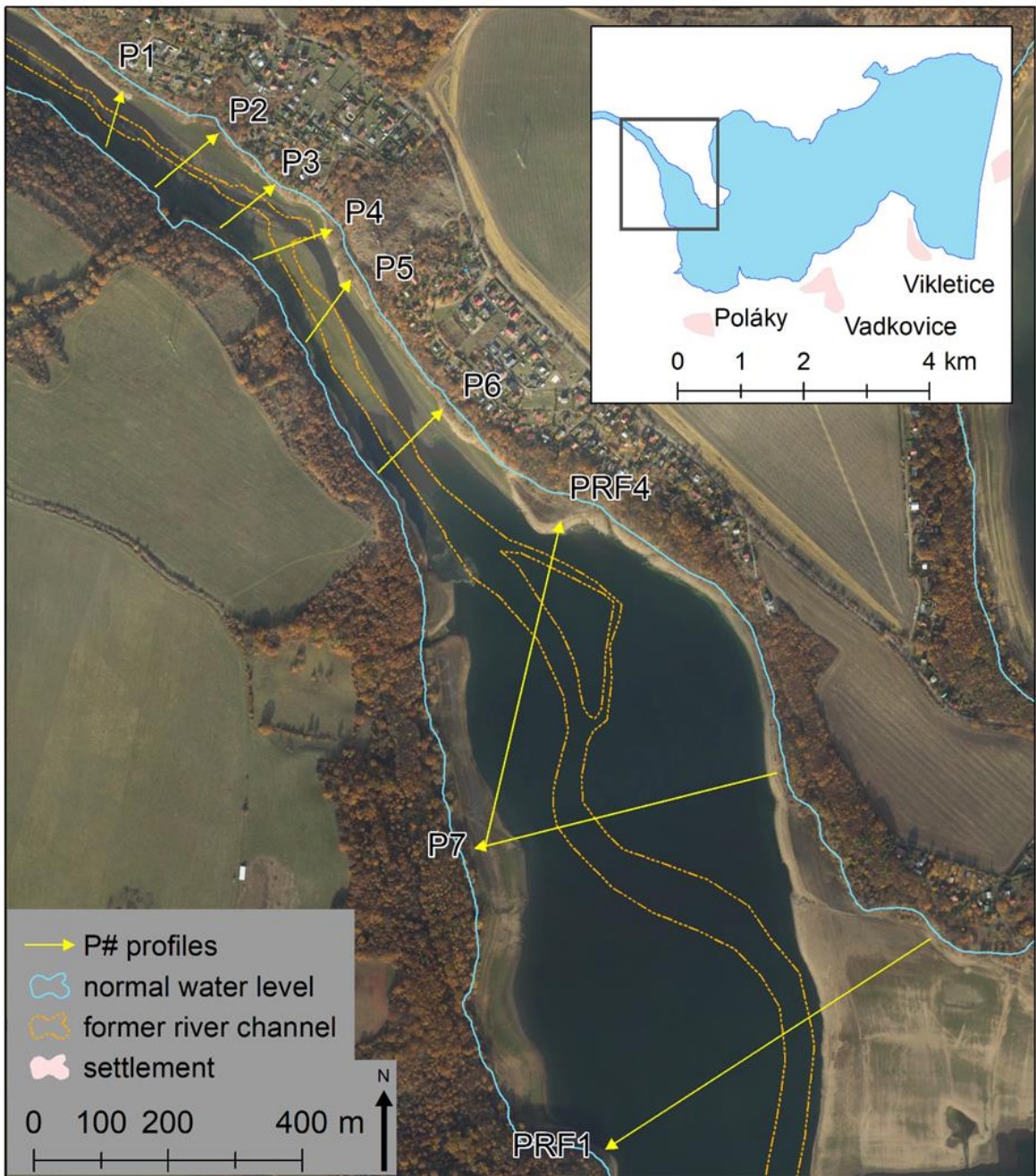

**Figure 3.** Detailed maps of the inflow to the Nechranice Reservoir with the positions of transect profiles (P#) for evaluation of elevation changes and profiler transects (PRF#). The orthophoto shows the 2019 drawdown.

## 2.2. Pre-Dam Topography Derived from Archival Aerial Photographs

Archival aerial photographs are used widely for landscape change analysis, archeological research, and georelief reconstruction because they contain information valid at the time of their acquisition [16,24,45]. Suitable processing of archival aerial images with sufficient overlap using the standard methods of photogrammetry or Structure from Motion modeling allows the user to derive not only detailed orthophotos but also a digital surface model (DSM) of the former landscape (the pre-dam topography in our case) [22,23,46].

For the area of the Czech Republic, archival aerial images are available in the national archives managed by the Military Geographical and Hydro-meteorological Institute in Dobruška (VGHMÚř) or

the Czech Office for Surveying and Cadaster (ČÚZK). Most of the archival aerial images are already available in the digital form or can be scanned on demand with a professional photogrammetric scanner.

The current methods of processing archival aerial images allow, in most cases, the processing of DSMs of large areas [24]. However, it is difficult to process archival aerial imagery of areas with large landscape changes (water reservoirs and open-pit mining). The elements of the interior and exterior image orientation, including the catalogs of ground control points (GCPs) used for the original aerial image processing, are often missing [16,23]. The GCPs have to be identified in the contemporary data (usually orthophoto data) and on the archival aerial images. The elevation information is acquired from a field survey, LiDAR data, or (in an exceptional case) from archival topographical maps.

Detailed background research was performed in the ČÚZK and VGHMÚř archives, and several time series of archival aerial images for the Nechranice and Skalka reservoir inflows were identified. Seven aerial images from the year 1948 (approximate flight altitude of 4.2 km) were chosen for the inflow of the Skalka Reservoir, and 17 aerial images from the year 1963 (approximate flight altitude of 2.4 km) were selected for the inflow of the Nechranice Reservoir. These two time series were chosen based on data coverage, image overlap, and image quality. The archival aerial images for the areas of the studied reservoirs were processed in Agisoft Metashape (St. Petersburg, Russia).

Several data sources were used for identification of the GCPs required for the processing of the archival aerial images of the studied reservoirs: contemporary orthophotos, archival orthophotos from the 1950s processed by the State Environmental Fund of the Czech Republic, and available topographic maps.

### 2.3. Pre-Dam Topography Derived from Old Maps

The Skalka Reservoir is sufficiently covered by topographic maps (scale 1:5000) issued by the Central Geodesy and Cartography Administration in 1960. The contour interval of the maps is 1 m, with a 0.5 m interval in the flatlands. The map for the Nechranice Reservoir was obtained from the archive of the Ohře River state management (Povodí Ohře). This topographic map (scale 1:5000) was created prior to the dam construction with a contour interval of 1 m.

Processing of the old maps followed a common workflow. Digital copies of map sheets covering the Skalka Reservoir were georeferenced using projective transformation to the known coordinates of the map sheet corners. The Czech National Coordinate System S-JTSK (EPSG 5514) was used. The maps covering the Nechranice Reservoir were obtained in digital form and georeferenced into S-JTSK. The corresponding contour lines were hand digitized. Interpolation using Delaunay triangulation is suggested for this type of data [47].

### 2.4. Reservoir Bottom Topography

Within our areas of interest, bathymetry mapping with single-beam sonar was performed at the Nechranice Reservoir in July 2014, June 2017, and May 2019 and at the Skalka Reservoir in May 2018 and September 2019. GPR surveying was performed at Nechranice Reservoir in June 2017 and at Skalka Reservoir in September 2018 and additionally during a time of higher water level in July 2019. The sub-bottom profiler was used at Nechranice Reservoir in March 2019.

Drought periods during the last decade in combination with dam construction work led to extremely low water levels in the reservoirs and allowed us to survey the normally inundated areas with standard topography mapping tools. Photogrammetry was used in the Nechranice Reservoir in August and November 2018. The low stand of the Skalka Reservoir was surveyed using LiDAR in 2011 within the national survey carried out by the ČÚZK and in November 2019 by our survey.

#### 2.4.1. Sonar Survey

The sonar survey was conducted using a Humminbird Helix 7 CHIRP DI GPS G2 device equipped with standard GPS (Humminbird, Eufaula, AL, USA) placed on a small inflatable boat or paddleboard-based floating platform with an electric drive, as described in [44]. The raw data

were exported as single point coordinates [easting, northing, relative depth] by the Humminbird PC (Humminbird, Eufaula, AL, USA) software. The water level was measured before sonar data acquisition using an RTK GNSS South surveying system with an accuracy of approximately 3 cm for comparison with the data from other relevant processed data sources (the pre-dam topography, aerial or LiDAR survey). The hydrographic survey used for compiling nautical charts is performed in accordance to the "Standards for Hydrographic Surveys (S-44)" issued by the International Hydrographic Organization [48]. Our bathymetric survey was not focused on detailed survey of the whole reservoir bottom, and thus the survey pattern (lines) was defined with respect to the size of the reservoir and the technical possibilities of the survey equipment (battery life). The survey line distance within the deltas of the Skalka and Nechranice reservoirs was approximately 50 m.

The survey in the Nechranice Reservoir in summer 2014 (water level 266.75 m) and 2017 (water level 267.377 m) covered the standard water level. The survey in spring 2019 (water level 267.82 m) was performed to document the changes after the extreme low water level in autumn 2018 (water level 261.74 m).

### 2.4.2. GPR and Sub-Bottom Profiler

In this study, the TerraSirch 3000 (GSSI, US) GPR unit operated from an inflatable boat was used with 200 and 400 MHz antennas towed in another boat at constant speed. A series of transverse GPR sections across the Skalka Reservoir deltas (Figure 2) was measured. The dielectric constant was commonly set at 80 for fresh water [49,50] and the GPR data were filtered by dewow, static correction, background removal, band-pass filter, and manual gain using ReflexW software (Sandmeier, Germany). The maximum water depth penetration was approximately 5 m. The water depth conversion from GPR two-way travel time to altitude was performed using the bathymetry data from the Ohře River state management. In total, 18 GPR profiles (153 to 567 m long) were measured at the Skalka Reservoir.

In this study, the narrow beam parametric sub-bottom profiler SES-2000 compact (Innomar Technologie GmbH, Rostock, Germany) with dual-frequency was used in the Nechranice Reservoir (Figure 3). The instrument consists of a topside unit and a sub-bottom transducer operated from an electrically powered boat. The instrument operates with dual frequency, a primary high frequency (85–115 kHz) suitable for bathymetry surveying, and a secondary low frequency (2–22 kHz) for sub-bottom penetration. The water depth penetration range is 0.5 to 400 m, the layer resolution is 1 to 5 cm, and the maximum sediment penetration is 40 m, depending on sediment type and noise. For displaying the vertical scale of the sub-bottom profiler (m), the sound velocity of 1500 m s$^{-1}$ was used.

### 2.4.3. Photogrammetric Survey in the Nechranice Reservoir

The construction works associated with the lowering of the water level at the Nechranice Dam were announced in spring 2018. The announced water level was achieved in summer 2018. A photogrammetric survey was performed to check whether the newly exposed areas correspond with sediment sampling sites [8] and potentially to compare the data with results of the sonar survey. Unfortunately, the water level was not low enough to allow these actions. However, the extremely dry summer and autumn of 2018 in central Europe caused the water level in the Nechranice Reservoir to decrease far beyond the announced level. Thus, a second aerial survey was performed in November 2018.

The aerial survey was performed using a small aircraft. A Hasselblad A6D-100 camera with a Hasselblad HC f3.5/50 mm-II lens was used. The surveyed area in November 2018 was 30.6 km$^2$, the average flight altitude was 780 m, the ground resolution 6.8 cm/pix, and the total number of images was 1605.

### 2.4.4. LiDAR Survey in Skalka Reservoir

The water level in the Skalka Reservoir was lowered in summer 2011 due to construction works on the dam. The national aerial survey, carried out by ČÚZK in this region, was performed at the same time. The LiDAR data, called the Digital Terrain Model of the Czech Republic of the 5th Generation (DTM 5G), are available for the whole Czech Republic as an irregular set of points, with a 1 point/m$^2$ average density. The root mean square error (RMSE) of the elevation information is 18 cm in open areas and 30 cm in areas covered by vegetation. The points represent the bare ground [51].

A second huge discharge of the Skalka Reservoir was identified on Sentinel-2 satellite imagery in November 2019. The Riegl VUX1-LR scanner in combination with a small aircraft was used for the LiDAR survey (18 November 2019). The scanner is equipped with the Applanix IMU unit (recording the yaw, pitch, and roll angles) and a precise RTK GNSS receiver. The following parameters were used: flight altitude of 300 m, scan stripes with 50% overlap, strip distance of 200 m, scanner Field of View80°, pulse repetition rate of 400 kHz, and scan rate of 48 lps. The resulting point density on the exposed dam bottom was 25 points/m$^2$. Raw data processing was performed in PosPac (trajectory data postprocessing using the national network of GNSS reference stations) and RiProcess (point cloud processing).

### 2.5. Quality of Input Data and Survey Methods

The reservoir bottom dynamics can be studied based on the introduced archival spatial data and the spatial data collection methods. However, quality assessment of the collected data is an important part of the research because it can influence the results [13,27,47]. Quality assessment is usually performed on a set of control points within the area of interest. Nonetheless, evaluating the quality of data used for reservoir bottom dynamics/change analysis is problematic. There is no relevant (directly measured) set of data to be used for quality assessment of the historical data.

The historical data (old maps) used for the pre-dam bottom modeling are a unique data source incomparable with the precise data acquired using modern data collection techniques. We can compare areas around the reservoir, which have not changed since the reservoir inundation, but we are unable to verify the quality of the elevation data at the bottom of the reservoir.

Verifying the quality of current spatial data used for dam bottom modeling is also problematic in the case of large and deep reservoirs. The quality of sonar-based bathymetric mapping may be evaluated under laboratory conditions or by comparison with measured profiles [52]. A laboratory test was performed in the summer of 2014, prior to the first bathymetric survey of the Nechranice Reservoir, in a shallow pool. The results did not indicate any significant differences from the directly measured depth (1.3 m). Direct and systematic measurements of profiles/transects within the reservoirs, with respect to the size, depth, and nature of the unconsolidated bottom mud, are difficult to carry out.

The LiDAR survey performed in 2011 by ČÚZK in the Skalka Reservoir has a vendor-defined accuracy [51] of RMSE of 18 cm in open areas. The Riegl VUX1-LR, used for the survey of the Skalka Reservoir in autumn 2019, has an accuracy of laser beam of 15 mm and a precision of 10 mm. The equipped Trimble AP20 GNSS-inertial system has a defined (postprocessed) position accuracy (RMS error) of 0.02–0.05 m. The trajectory was postprocessed in PosPac software with 5 cm positional accuracy.

In the GPR profiles, to calculate the accurate depth axis from GPR two-way travel time, the propagation velocity of 0.0335 m ns$^{-1}$ [50] was used together with several test scans. The resulting relative depth was converted to absolute elevation using the actual water level values.

## 3. Results

### 3.1. Precision of Modeling the Pre-Dam Topography

The pre-dam topography was modeled based on two types of data—archival maps (containing contour lines) and archival aerial imagery.

In total, 7 archival aerial images from 1948 were processed for the Skalka Reservoir, and 17 archival images from 1963 were processed for the Nechranice Reservoir; they were processed in Agisoft Metashape (v. 1.5.0). Precise definition of the GCPs is important for the processing of archival aerial imagery and for the creation of a georeferenced orthophoto and DSM. The extensive landscape change (dam construction and open-pit mining) in the area of the Nechranice Reservoir [53] made it impossible to identify a sufficient number of GCPs with the corresponding quality. The DSM for the Nechranice Reservoir inflow derived based on the 1963 imagery is randomly overestimated or underestimated by several meters in comparison with the DEM derived from the archival topographic map.

Archival aerial photographs contain a lot of grain noise and many artifacts caused by their age. The grain noise and the various scratches influence the quality of the resulting DSM. The methods for noise removal described by [23] were applied to the 1948 aerial images covering the Skalka Reservoir. The result was a slightly smoother DSM that was still too noisy to be used for detailed pre-dam topography analysis.

The old maps were used to model the pre-dam topography. The contour lines, elevation points, and edges (in the case of the Skalka Reservoir) were used as the interpolation input. Use of the Triangulated Irregular Network (TIN) was proposed by [47] for this type of data and purpose. The resulting TIN was further converted to a grid with a spatial resolution of 1 m/pix.

A total of 742 points (derived from the digitized contour lines) in the area surrounding the Skalka Reservoir was selected for the quality assessment of the digitized contour data. The points were located in areas where no landscape change was expected (fields, forested areas, and steep and flat slopes). From the 742 points, 113 points located in flat land (simulating the reservoir bottom) were selected for the second accuracy analysis. The data were compared with the LiDAR data DTM 5G. The RMSE values of the differences between the two tested datasets are presented in Table 1.

The same approach was used in the Nechranice Reservoir. The points were selected around the inflow and in the southern part of the reservoir because the landscape in the northern part had changed with open-pit mining (reclamation sites). In total, 358 points in the flatlands and 558 points in the steep slopes around the reservoir inflow were selected for the quality analysis. Based on the large RMSE (2.8 m) in the inflow area (steep and forested slopes), the contour lines were not used for the analysis of the bottom morphological dynamics. Thus, the inflow-delta bottom analysis was based only on the verified benchmark points identified in the processed map. The RMSE of the elevation differences in the flatlands in the southern part of the Nechranice Reservoir was 68 cm (Table 1).

## 3.2. Precision of Modeling the Reservoir Bottom Topography

A sonar bathymetric survey was carried out in the Skalka and Nechranice reservoirs. As a result of the size of the reservoirs and the technology used (single-beam sonar), the collected data points were sparse. The quality of the surveyed data was checked in the Nechranice Reservoir during the dry period. Sixteen points were surveyed with RTK GNSS at a previously paved road inundated under standard conditions. Three sonar survey transects crossed this road, so only three points could be compared in this way. The RMSE of the differences between the sonar and RTK GNSS-based altitudes was 9.5 cm.

Another (but less accurate) way of comparing the bathymetric data is to compare the bathymetry data (points) with the DSM of the bare bottom derived from aerial imagery (see Table 1). This quality assessment is influenced by possible sediment transport and DSM accuracy. Therefore, the RMSE values are only approximate. Altogether, 1122 points, excluding the inflow delta, were sonar surveyed (in 2017) in areas exposed in autumn 2018 during the extremely low water level. The RMSE of the elevation differences was 69 cm. Furthermore, 34 points were sonar surveyed on formerly paved surfaces (identified by the sonar as "hard bottom"), and the RMSE of the elevation differences was 24 cm.

Comparison of the sonar survey and the GPR was possible only at parts of P15 and P16 at the Skalka Reservoir, because the two surveys intersected only in these parts. The RMSE of the elevation differences was computed based on 254 points (Table 1).

**Table 1.** Quality evaluation of selected methods. DSM: digital surface model, RMSE: root mean square error.

| Measurement Type | Number of Control Points | Elevation RMSE |
|---|---|---|
| Archival maps—Skalka | 742 | 55 cm |
| Archival maps—Skalka (flat land) | 113 | 30 cm |
| Archival maps—Nechranice (flat land) | 358 | 68 cm |
| Archival maps—Nechranice (steep slopes) | 559 | 2.8 m |
| Sonar bathymetry | 3 (RTK measured CPP) | 9.5 cm |
| Sonar bathymetry | 1122 (all points on aerial DSM) | 69 cm |
| Sonar bathymetry | 34 (points on paved areas) | 24 cm |
| GPR survey | 254 (sonar surveyed) | 14 cm |
| Aerial photogrammetry | 16 | 4 cm |
| LiDAR—2011 | - | 18 cm (defined by data provider) |
| LiDAR—2019 | 154 | 9.5 cm |

The sonar-surveyed points describing the reservoir bottom topography were interpolated into a DEM in ArcGIS PRO using the Topo To Raster function. A spatial resolution of 1 m was used for the Nechranice Reservoir inflow delta, and a spatial resolution of 5 m was used for the whole reservoirs. DEM interpolation based on such a sparse dataset is influenced by many interpolation artifacts. Thus, the dam-bottom dynamics were further studied in transects. In the Nechranice Reservoir, to minimize the interpolation artifacts on the resulting transects, the transects were defined in areas well covered by the sonar data and in directions of the sonar survey. The transects in the Skalka Reservoir were defined by the GPR survey; thus, the sonar-based data were not used for the analysis.

The quality of the photogrammetric survey used in this study in the Nechranice Reservoir may have been influenced by the number and distribution of GCPs within the surveyed area [21]. Altogether, 15 GCPs were surveyed around the water reservoir. The aerial survey had to be performed on a day-to-day basis (weather conditions, aircraft hiring possibilities), and the GCPs were thus surveyed with RTK-GNSS several days after the aerial survey. The objects identified on the aerial images were used as the GCPs (i.e., road markings, sharp edges on sidewalks). The aerial images were processed in Agisoft Metashape (v. 1.5.0). The DSM was derived with a spatial resolution of 27 cm/pix and used for the transect analysis in the Nechranice Reservoir.

The GCPs were regularly distributed around the reservoir, but there was no GCP within the reservoir area (13.4 km$^2$) or in the surrounding forested or inaccessible areas. To check whether the "gap" in the GCPs affected the resulting DSM in any way, 16 control points measured on the formerly paved road were used. The RMSE of the altitude differences of 4 cm corresponds to the processing accuracy of this method.

The DEM resulting from our LiDAR survey was exported with 1 m/pix spatial resolution (with filtered vegetation) to correspond with the 1 m/pix resolution of the DTM 5G. Our LiDAR survey was focused mainly on the emerged reservoir bottom, where it was impossible to measure any control points because of the unconsolidated mud surface. The quality of the LiDAR system was checked on an adjacent area with 154 RTK GNSS measured control points. The control points were measured with an accuracy of approximately 3–8 cm. The resulting elevation RMSE of 9.5 cm was a consequence of the nature of the collected dataset (no GCPs were used for LiDAR data processing as no GCPs could be measured in the Skalka Reservoir bottom) and the fact that part of the area was forested (it influenced both the LiDAR responses and the RTK GNSS measurements). The resulting DEM was used for the analysis of the bottom dynamics in the Skalka Reservoir.

### 3.3. Inflow to Skalka Reservoir

The strong reflector of the dam bottom can be seen in all GPR profiles up to a depth of 4 m (Figure 4). The bottom morphology in the GPR images is relatively flat with several deeply incised channels. The signal penetration in the dam bottom is relatively shallow and depends on the nature of the sediment and the depth of the water. From the GPR profiles, two main GPR facies with distinct reflection properties can be distinguished. The parallel and subparallel facies producing horizontal reflections are assigned to muddy reservoir sediment, as in [4] is interpreted as parallel bedding and is present mainly in the flat parts of the dam bottom and at the bottom of some channels (P14, Figure 4C), where aggradation can occur. A strong attenuation of the GPR signal is typical for these facies. The mound facies shows hyperbolic reflections and is assigned as sandy or gravelly channel sediments according to [4]. The mound structures are present in some parts of the dam bottom, such as in the old fluvial structures underlying the parallel facies (P6 and P14, Figure 4A,C, respectively). They are also found in the depositional banks of submerged channels, where the lateral deposition of coarse sediments can be expected (Figure 4A around 310 m).

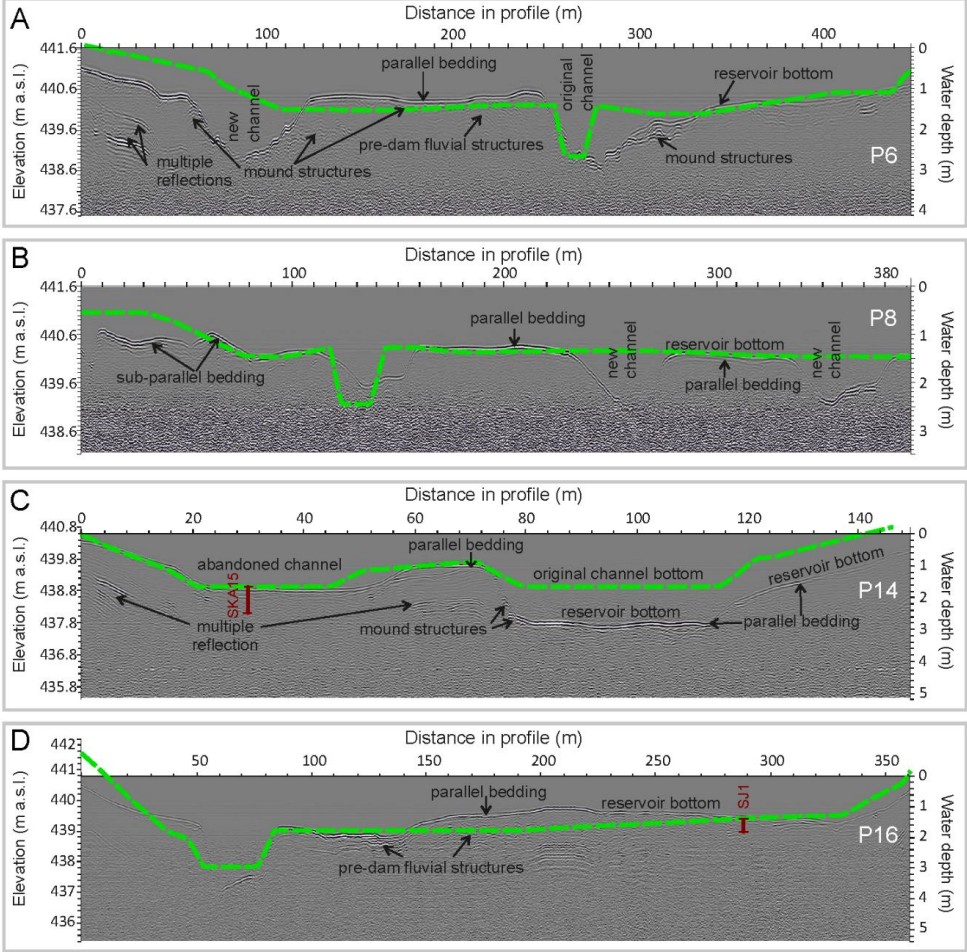

**Figure 4.** Ground-penetrating radar (GPR) images of selected transect profiles in the Skalka inflow. The positions of profiles are shown in Figure 2. (**A**) Transect profile P6, (**B**) transect profile P8, (**C**) transect profile P14, (**D**) transect profile P16.

The major difference between the recent DEM and the pre-dam topography is the formation of novel channels (Figures 2 and 4). Although the pre-dam river had a meandering channel, the damming produced a complex multichannel system (Figure 2) with the deposition on flat surfaces between these channels. The changes in the valley bottom topography at the spatial scale of individual profiles

through the reservoir inflow after damming were visualized by the differences between the recent reservoir bottom obtained by GPR and the pre-dam topography (Figure 5). Net differences (Figure 5, empty squares) were calculated as the average of the elevation differences for individual profiles, i.e., they correspond to the net elevation changes of the entire profiles. Net erosion was largest in P12 and P13, which were located in the narrowest part of the inflow. Net aggradation (Figure 5, full squares) was highest in the downstream profile P18, i.e., the profile that did not emerge during usual seasonal drawdowns. Neither was observed in the central part of the reservoir, but reservoir mud deposition was observed in fieldwork (unpublished results). Generally, the inflow profiles indicate net erosion rather than net aggradation. The means of the absolute values of elevation differences show overall changes in topography, including sediment reworking, at short distances. Interestingly, these overall changes in elevation were similar in all profiles, with a local maximum in the narrowest part of the inflow (profile P13, Figure 5).

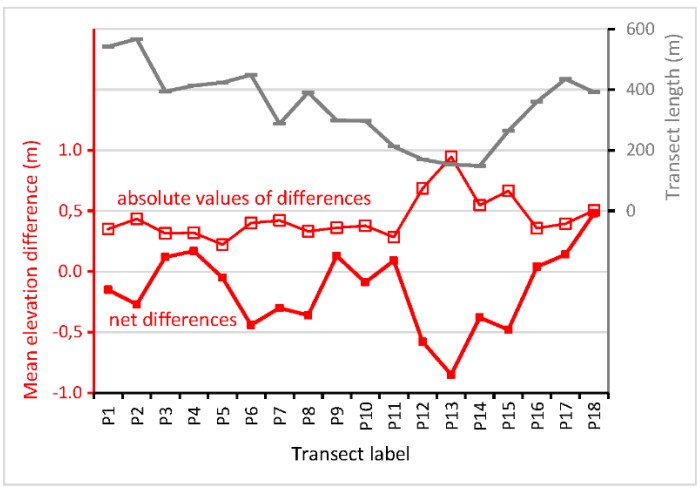

**Figure 5.** Downstream change in overall elevation difference in entire transect profiles in the Skalka inflow.

The geomorphic development in profiles P1 to P6 was characterized by prevailing lateral erosion in the former main channel (except for P3, Figure 6A), the incision of novel channels into pre-dam topography, and accumulation (aggradation) mostly in flat parts of the bottom distal to channels (Figure 6A–C). In-channel aggradation was observed only in P3; however, newly formed channels were larger than the original one (Figure 6A). Profiles P6 to P8 showed net erosion (Figure 5, Figure 6B). P9 to P11 showed minimal net elevation change (Figure 5), with the example of P10 shown in Figure 6C.

The bottom erosion was enhanced in profiles P12 to P15 situated in the narrowest part of the inflow (Figure 2), especially, the incision in the former channel (all profiles, example in Figure 6D) and the formation of novel channels (P12, P13, P15). In P16 to P18, the in-channel erosion was less intensive and accumulation in the flat part of the bottom distant from channels (Figure 6E) increased in the downstream direction, producing net aggradation (Figure 5).

Sediment cores were examined in profiles P10, P14, and P17. In P10, only the top 15 cm of the core contained Zn-contaminated sediments, which is in agreement with the negligible aggradation at the coring site after the damming (Figure 6C). In P14, the core was located in a channel that had already been abandoned before the river damming (Figure 2), and its entire thickness was contaminated by Zn (Figure 6D). In P17, the thickness of the sediment retrieved by coring corresponded to the thickness of the post-dam deposits, and the entire cored sediment was contaminated by Zn (Figure 6E).

The recent bottom dynamics was evaluated from LiDAR images for 2011 and 2019, which were both obtained during reservoir drawdowns, and GPR images of 2018 or 2019 (before the 2019 drawdown) (Figure 6). These recent changes were mostly only slightly above the range of uncertainty of the measurements. The downstream profiles showed LiDAR elevations (2011, 2019) in the floor distal

to the channel that were higher than the GPR result (2018) by ca. 15 cm (P13), 20 cm (P14, P15, P16), and 10 cm (P17), indicating that there may have been some aggradation during the drawdowns. The similarity of the LiDAR images of 2011 and 2019 documents that the most dramatic changes from the pre-dam topography occurred between the time of dam construction and 2011.

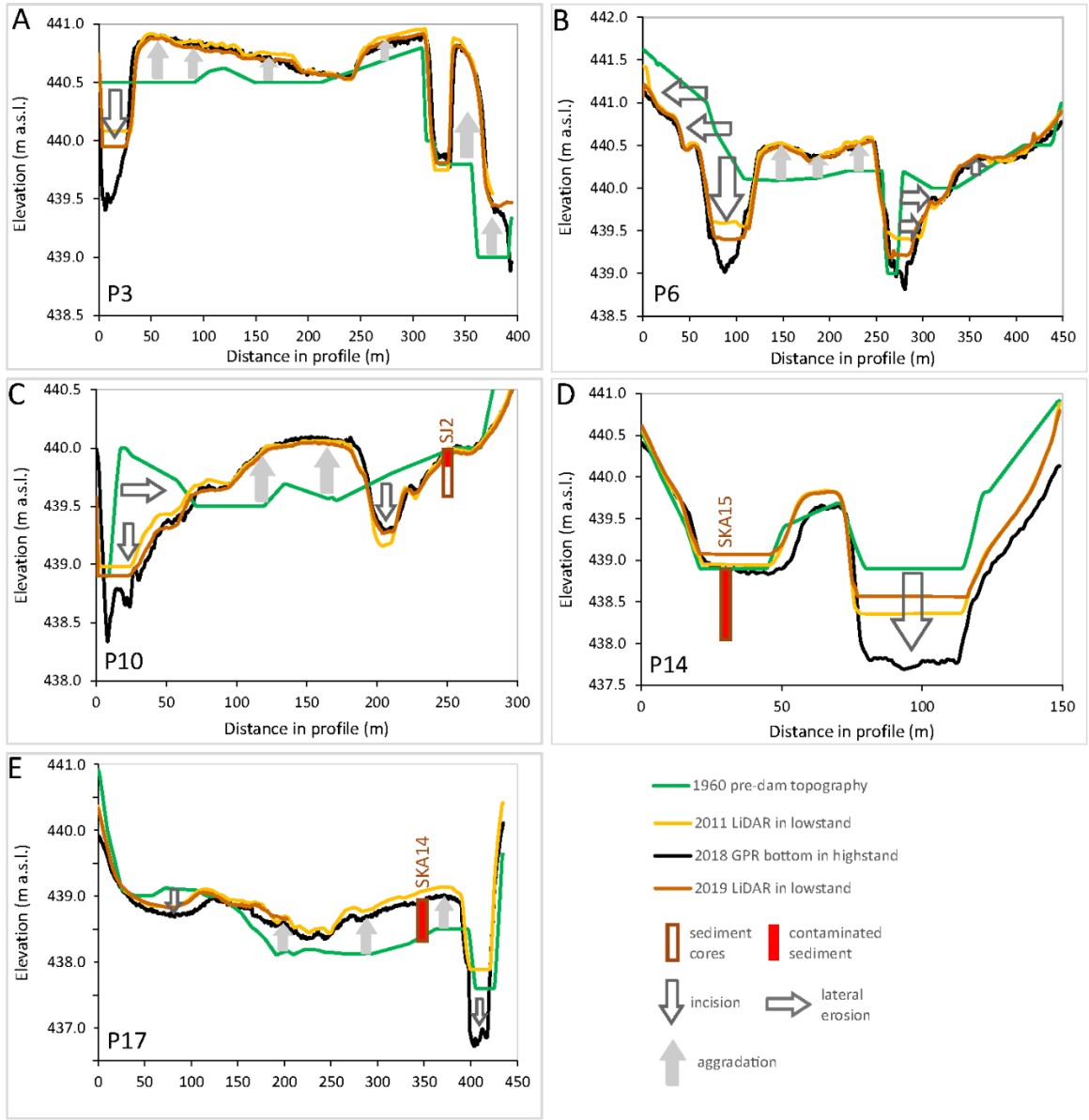

**Figure 6.** Elevation changes in selected transect profiles in the Skalka inflow showing changes within each profile. (**A**) Profile P3, (**B**) profile P6, (**C**) profile P10, (**D**) profile P14, (**E**) profile P17.

### 3.4. Inflow to Nechranice Reservoir

The pre-dam valley floor showed a V-shape profile, with a bending river channel and practically no floodplain or similar flat areas near the river channel (Figure 7A). At the present time, the reservoir sediment in profiles P1 to P6 has accumulated to a nearly 13 m thick stratal sequence (Figure 7C) that fills and levels the valley floor to 260–262 m a.s.l. (Figure 7B). The flat top of the sediment body in the Nechranice inflow is terminated by an elevation decline from 261 to ca. 256 m a.s.l. (Figure 7B,D), which is a typical feature expected for the inflow delta, with no channel incised under the normal water level, as shown in the sonar bathymetry (Figure 7B), and with a flat delta top and steep delta

front (Figure 7D). The lack of sharp boundary between delta top and front is typical for reservoirs with fluctuating water levels [54]. The current channel is incised into the delta surface in a position that is very different from the pre-dam channel (Figure 3).

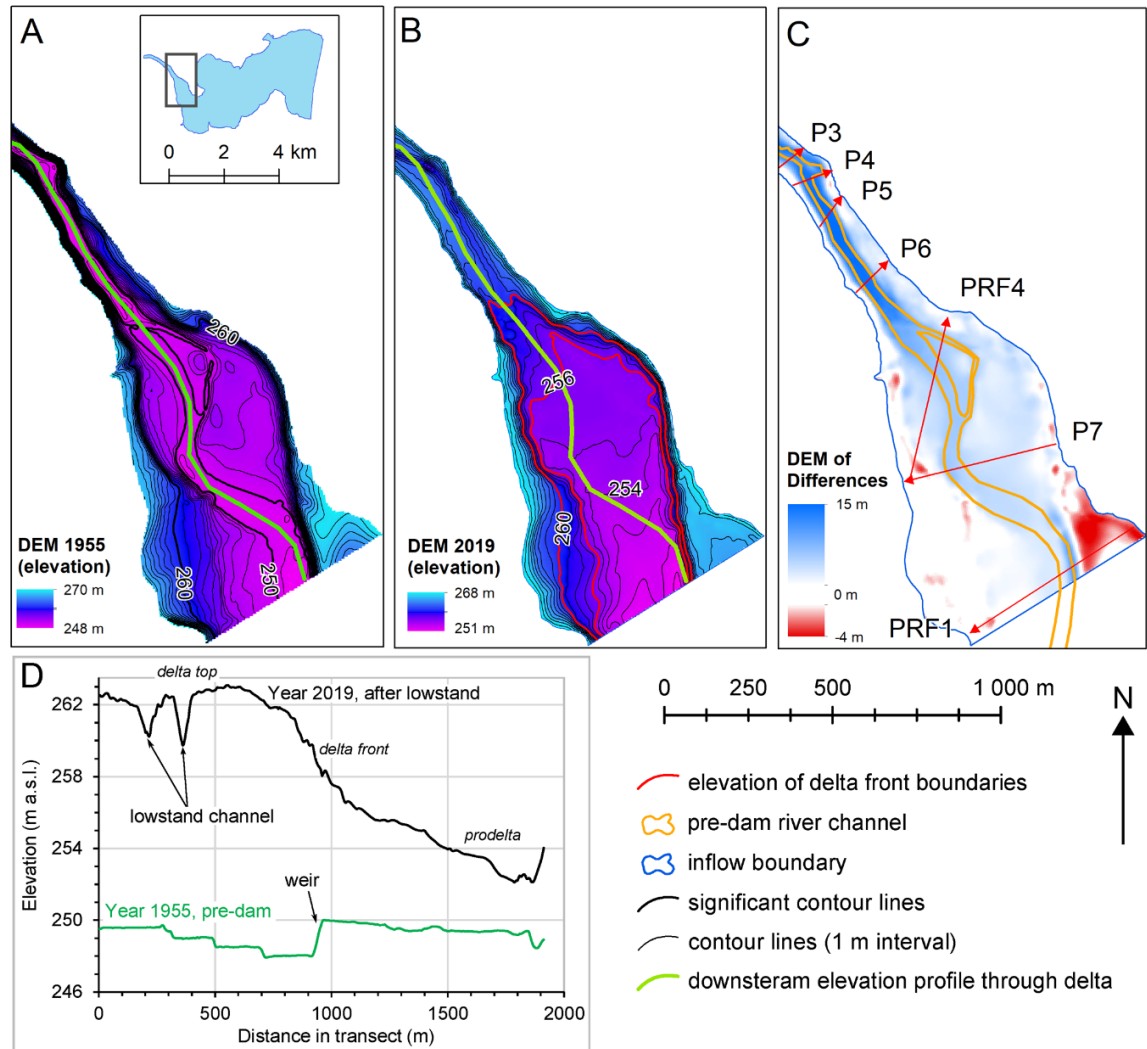

**Figure 7.** Map of elevation differences in the Nechranice inflow. (**A**) Pre-dam topography, (**B**) current topography from sonar imaging, (**C**) reservoir sediment thickness as an elevation difference of digital elevation models (DEMs) in panels A and B, (**D**) elevation change in the inflow delta along the pre-dam thalweg (green line in panels B and C) with an assignment of delta morphological features.

The water column in the Nechranice Reservoir inflow was too thick for the GPR device. Thus, the sub-bottom profiler was used with the goal of producing an image of the subsurface sediment structure. The surface of the reservoir bottom was clearly visible in all profiles at depths of up to 17 m. It formed a U-shaped valley (Figure 8), which was interpreted as the surface of the delta. In the flat parts of the bottom, the profiler reflection was accompanied by shallow horizontal reflectors representing only a thin subsurface layer of the vertically deposited reservoir sediments. A slump, probably from a collapsed submerged valley bank, was found in the GPR1 profile (Figure 8A). Bedding was not distinct in the profiler images (exceptions are shown in Figure 8B), which we attribute to the massive structure of the deltaic sediments.

The elevation changes of the Nechranice delta surface were evaluated from sonar imaging and aerial photographs (Figure 9). Their further description is related to the last two measurements performed in 2018 and 2019. All profiles from the delta top (P1 to P6) showed channel incision as deep as 2.5 to 3 m

in P2 to P4 and lateral erosion of the reservoir bottom (except for P2, Figure 9B), while the elevation of the bottom outside of the channel was nearly constant (except for P3, Figure 9C). Downstream profiles P5 to P7 showed lateral channel shifts with minor channel incision but with considerable lateral shifts and erosion (P5 and P6, Figure 9E,F). The profile in the delta front (P7, Figure 9G) showed aggradation and lateral deposition, which was probably through the accumulation of materials in the delta front from lateral erosion and channel incision in the delta top (P1 to P6).

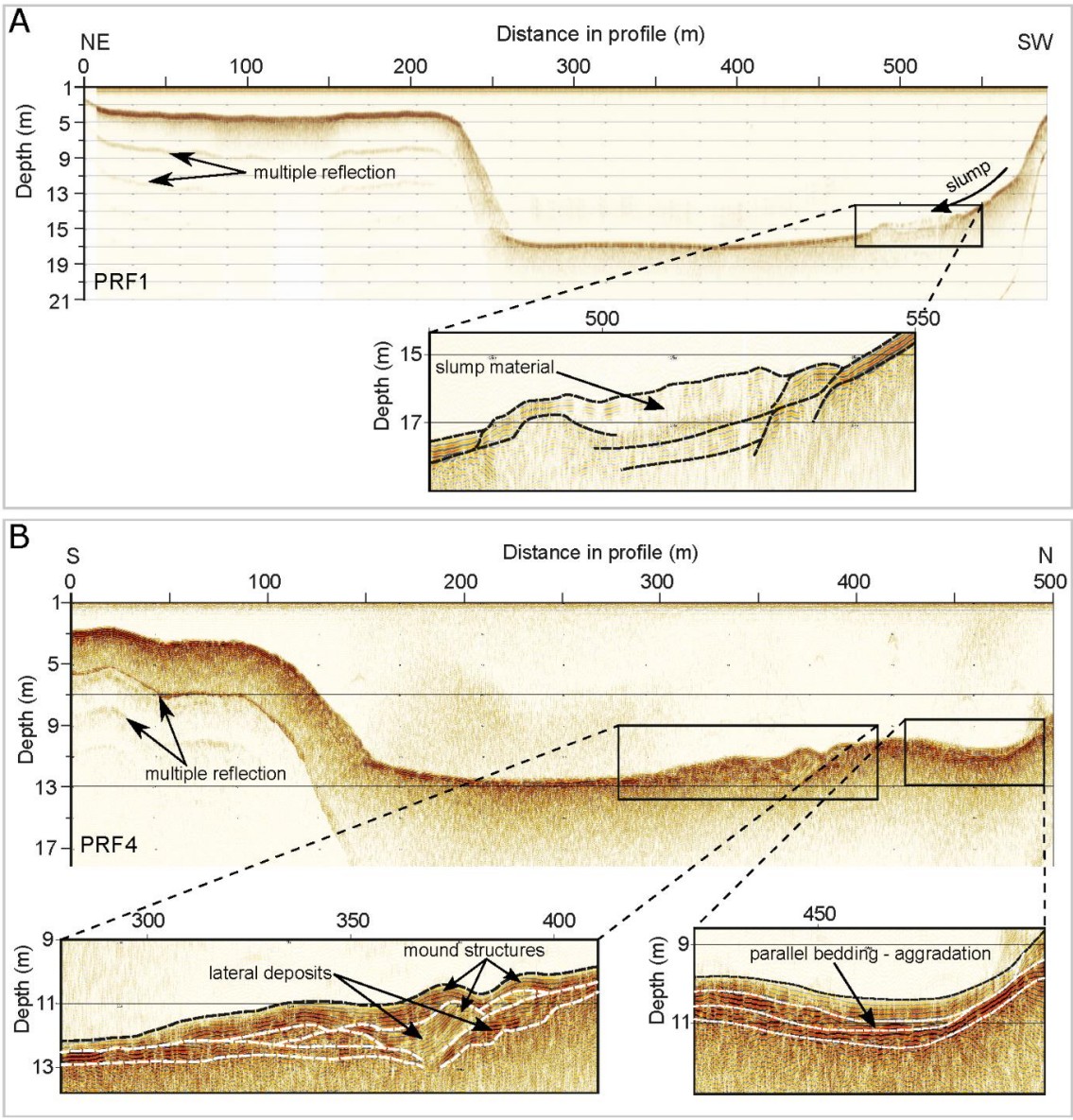

**Figure 8.** Profiler images in two selected profiles in the Nechranice inflow. (**A**) Profile PRF1, (**B**) profile PRF4.

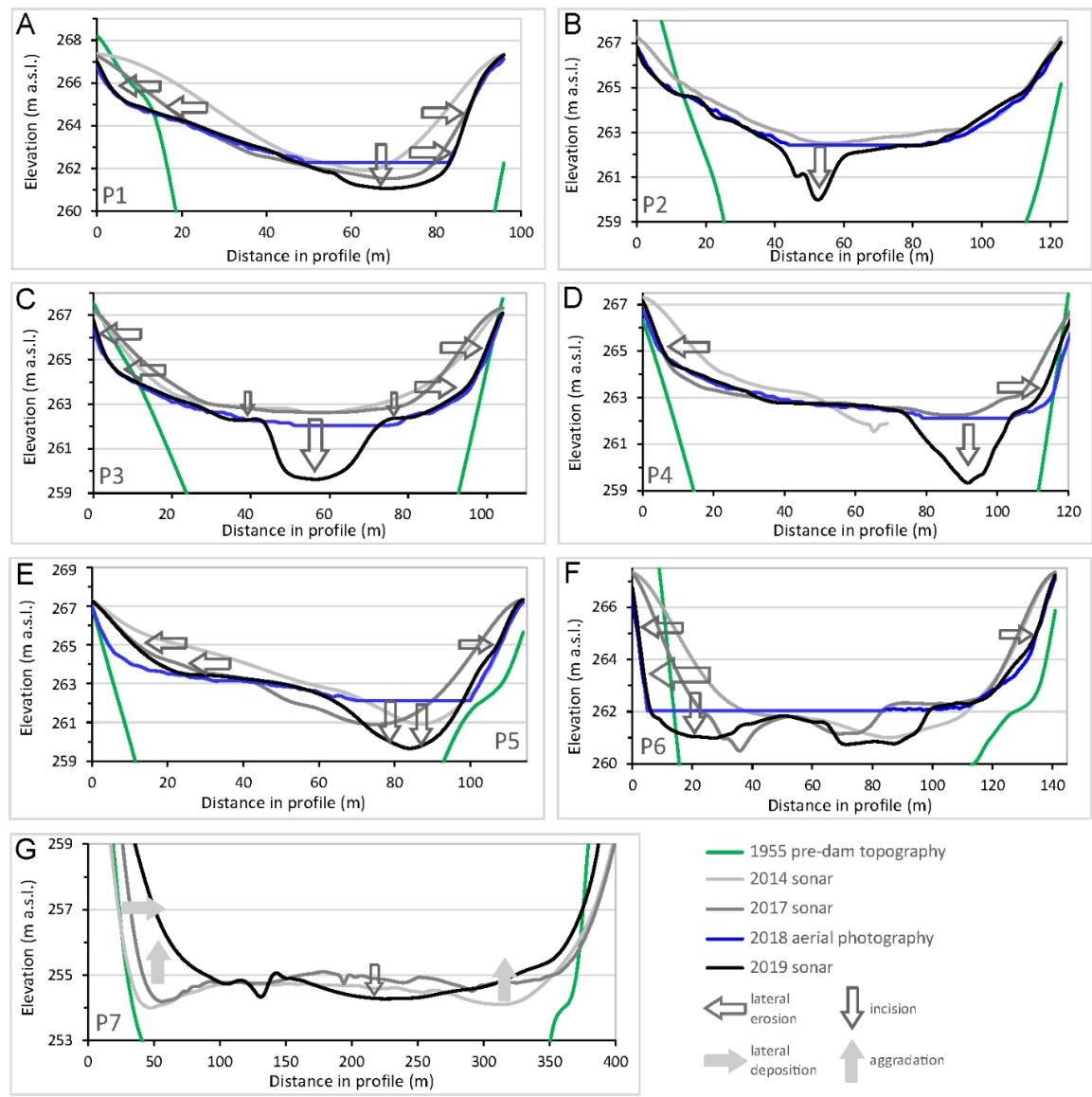

**Figure 9.** Elevation changes in selected profiles through the Nechranice inflow. (**A**) Profile P1, (**B**) profile P2, (**C**) profile P3, (**D**) profile P4, (**E**) profile P5, (**F**) profile P6, (**G**) profile P7.

## 4. Discussion

### 4.1. Data Quality, Evaluation of Routines/Techniques, and Verification of Results

The modeling of pre-dam topography is dependent on the spatial data available. Two types of historical datasets were tested for the two reservoirs of this study. Archival aerial photographs seem to be a reasonable data source for landscape reconstructions, and this type of data is used widely [16,23,24]. In this study, much attention was devoted to georelief reconstruction based on the archival aerial imagery because this information source would be very beneficial for the pre-dam surface modeling. The resulting DSM would also contain landscape forms mostly not retrieved from contour lines, such as sharp edges, cliffs, and vertical slopes, and the orthophoto processing could be rapid, well-described, and straightforward. The land cover could be reconstructed successfully based on the orthophotos derived from the source images, but several challenges occur in georelief reconstruction. The GCP catalogs used for the original image processing are often missing, and their replacement by objects identified on the archival aerial photographs and on the contemporary orthophotos brings uncertainty that propagates through the subsequent computation. In the case of large landscape changes, such as

the creation of the Nechranice Reservoir, it can even be impossible to identify a sufficient number of GCPs. The other factor that influences the quality of the derived DSM is the quality of the raw imagery. Sevara et al. (2017) described methods for minimizing the effects of grain and radiometric noises on the resulting DSM. However, application of the described methods to our raw imagery did not have the desired smoothing effect. This was probably caused by the images having different (lower) quality than those used by [23]. The noise contained in the resulting DSM and the overestimation and underestimation of the DSM caused by the GCP uncertainty would not matter in applications with lower accuracy requirements and at a smaller scale. However, in our case, when we need to compare the pre-dam topography with the current reservoir bottom at a larger scale, an inaccuracy of several meters is not suitable for the DEM differential analysis.

The second historical data source is old maps. The selection of accurate maps and precise pre-processing (georeferencing) is crucial for obtaining precise results. The area of the Skalka Reservoir was covered by detailed maps (scale 1:5000) created in 1960 by direct survey methods. Based on our background research, this is one of the most detailed and precise archival maps available for the former Czechoslovakia. The elevations obtained from the digitized contour lines were verified by the contemporary LiDAR data (in areas outside the reservoir). The RMSE of 30 cm of the differences in the open and flat areas (the global RMSE for the LiDAR data used was 18 cm in open areas) makes the pre-dam topography of the Skalka Reservoir very precise. A slightly different situation occurred for the Nechranice Reservoir. Detailed topographic maps (1:5000) were created only at selected areas near the "Iron Curtain" during the Cold War, and the Nechranice region was not covered by these maps. However, the map used for modeling the Nechranice Reservoir pre-dam topography also had a scale of 1:5000. The map contained two types of elevation information, contour lines, and elevation points. The elevation points are part of the geodetic river polygon and were measured precisely according to the national standards. The method used to create the contour lines is unknown. The accuracy of the map was again verified by comparison with the LiDAR data. In flat areas around the reservoir (simulating the flat reservoir bottom), the RMSE was 68 cm. In the steep slopes around the inflow, the RMSE was several meters. Thus, the pre-dam topography of the Nechranice Reservoir was reconstructed based on digitized contour lines (in the reservoir bottom) and measured elevation points (in the reservoir inflow). The steep slopes in the inflow were further omitted from the analysis of the inflow profiles.

The bathymetric survey represents a very common method of reservoir bottom mapping. In the case of surveying large reservoirs, it is difficult to verify the data quality. During the drawdown at the Nechranice Reservoir in 2018, several control points were surveyed with RTK GNSS, and the whole reservoir was covered with photogrammetric imagery. Then, the accuracy of the sonar survey was verified using three directly measured points. The RMSE of 9.5 cm corresponded to our expectations. The bathymetric data were also compared with the photogrammetry-based DSM, and the RMSE of 24 cm (on hard surfaces) was influenced by sediment transport during the water runoff. The single-beam sonar used for the survey produces (in comparison with the size of the reservoir) sparse data. Thus, the profiles analyzed in the Nechranice Reservoir were defined in areas sufficiently covered with bathymetric data.

GPR was used in both reservoirs as it offers another view of sedimentation processes at the reservoir bottoms. One of our aims was to compare and verify the GPR data (the "bottom response") with a sonar-based bathymetric survey. The GPR survey for both reservoirs was also planned to acquire information on sub-bottom sedimentary structures. However, the depth of the water in the Nechranice Reservoir inflow was too large for the GPR antenna. The GPR profiles in the Skalka Reservoir were selected as different profiles from those of the bathymetric survey. Only parts of the two GPR profiles (P15 and P16) were identical to the bathymetric survey performed in the spring of 2018. Altogether, 254 of the points surveyed with sonar could be compared to the GPR survey. An 8 cm RMSE of the elevation differences in P15 was computed based on 126 points. An RMSE of 18 cm (computed based on 128 points) in P16 was influenced by sediment reworking. The total RMSE of 14 cm (for both P15 and P16) demonstrates the high quality of the surveyed data. Then, we found that

the topography of the Skalka inflow bottom was too complex spatially, which made the interpolation between sonar-based datasets and GPR profiles too rough. Thus, the sonar results were omitted in the evaluation of results.

Due to the range of methods used for the reservoir bottom survey, we had a choice of different data and were not limited by the potential uncertainty in the obtained datasets.

With respect to the low water level, we had the opportunity to map the bottom of the reservoirs using methods normally employed for mapping the landscape, i.e., photogrammetry and LiDAR. These methods, producing dense point clouds, were used not only for modeling the normally submerged reservoir bottom but also for control datasets for analyses of the previously submerged bottom. The quality of the surveyed data was also verified. Photogrammetric modeling requires regular spacing of GCPs over the surveyed area. These conditions could not be fulfilled in the reservoir bottom, because it was impossible to define GCPs there. A comparison with the RTK GNSS control points, surveyed in the middle of the reservoir (inundated under normal conditions), resulted in an RMSE of 4.5 cm, which proves the high accuracy of the obtained DSM.

The LiDAR survey carried out in the Skalka Reservoir helped us to identify submerged active channels in detail. The quality of the dataset was verified in an adjacent location (as no control points were measured within the reservoir). The RMSE of 9.5 cm (Table 1) was perfectly suited for our study; however, it could be improved further by adding GCPs into the survey.

The data collected with sonar, GPR, LiDAR and aerial survey are available as Appendix A [55].

## 4.2. Inflow Sediment Dynamics

A delta sediment body has formed in the inflow to the Nechranice Reservoir (Figure 7D). Nechranice is the type of lake [54] with still big capacity to store sediment and prograding delta. The Nechranice delta top strata became emerged during the 2019 drawdown (Figure 3), which provided sufficient datasets for the discussion of delta body dynamics. The phenomenon of gravitational redeposition of fine reservoir muds, assisted by winds, currents, and water level lowering, was recognized about two decades ago and documented in a review paper by [56]. According to examples shown in the review paper by [1], the top of inflow deltas can be subjected to erosion by (a network of) newly formed incising channels on reservoir drawdowns. Those processes are obviously under-investigated, as follows from the lack of detailed case studies dealing systematically with the phenomenon. The impact of sediment sluicing (flushing) by elevated discharge on the bottom topography was discussed by [2] and [5]; however, their datasets did not exhibit resolution comparable to the resolution we obtained for this paper. Our results showed only a negligible net erosion in the reservoir inflows after the sluicing operation, i.e., the opposite of the result expected from the aforementioned reviews. Most of the papers describing the course of sediment sluicing from reservoirs have paid attention to the overall budget of the process and the downstream consequences (e.g. [2]) rather than its geomorphic manifestation in the reservoir bottom.

In the Nechranice inflow delta, the drawdown channel in autumn 2019 (water level 261.7 m) was incised 2.5–3 m deep into the top of the delta deposits (P2 to P4, Figure 9B–D). The total thickness of the delta deposits in those profiles was 12.7–13.6 m, which had accumulated during the 51 years of the reservoir's existence. Assuming a linear net deposition rate of the body, the incision reworked deposits of the last 9–12 years in approximately 1/10 of the delta width. The preceding drawdown occurred in October to November 2015 (water level 264.6 m a.s.l.), but its possible erosive channels, if channels had formed at all, were filled by sediment in subsequent years, as no corresponding depressions were found in 2018 and 2019 sonar images (Figure 9). It documents a considerable reworking of the upper strata in the delta sediment body, with the sediments transferred farther to the reservoir basin. The concentrations of Cu, Pb, and Zn contamination are 30–70% larger in the reservoir inflow than in the reservoir basin due to preferential deposition of contaminants in the delta [8], and thus sediment reworking and downstream transport are not desired.

An inflow delta *sensu stricto* has not formed in the Skalka Reservoir, although we expected its existence before fieldwork and actually intended to study it. Delta developments are limited in shallow reservoirs types according to Łajczak [54]. The overall elevation change after damming has been rather minor in the area of the Skalka inflow and highly variable spatially, with incision in the former main channel, the formation of numerous novel channels by incision into the pre-dam topography (Figure 2), and aggradation mainly in the reservoir floor distal to the channels (Figure 6). This pattern resembles the situation in the Seč Reservoir in the Chrudimka River, where isolated patches of reservoir sediments formed in the inflow and other material, temporarily stored here, was partly transferred further downstream in seasonal drawdowns [44]. This phenomenon of stepwise sediment transfer to deeper parts of a water body was discussed by [11] and [56], who denoted it "sediment focusing". Although a delta has not formed in the Skalka Reservoir, erosion and sediment reworking have been enhanced (as shown in Results), and the channel network was extended with enhanced local incision, as in the case of the Nechranice Reservoir, during the drawdown (Figure 2). The Seč and Skalka reservoirs have similar settings in the inflow part: a formerly broad floodplain with a low-gradient meandering channel. The maximal historic Cu, Ni, and Zn pollution in the Seč Reservoir was 10–60% larger in the inflow than in the basin. We have not mapped contaminant concentrations in the Skalka Reservoir; however, it is definite that in this shallow water body, sediments and contaminants are transferred directly downstream to the basin. For this reason, the Skalka Reservoir has limited potential performance for future pollution trapping, as was also concluded by Hošek et al. [42]. The local erosion of the Skalka inflow area compared with the pre-dam condition could have resulted from the lack of plant cover on the bottom compared to the dense pre-dam meadow vegetation. The reservoir inflows are free of plant cover in all reservoirs studied thus far, and their surface is thus susceptible to erosion. Enhanced lateral channel instability coupled with upstream increasing erosion and reworking of sediments including undercutting erosion banks was also observed in the inflow part of reservoirs in the Polish Carpathians [43,57].

The sediment focusing and downstream transport of the inflow deltas (including historical contamination) will have increasing importance. Seasonal droughts caused by less snow cover in river headwaters, and spring and summer droughts are among the features expected to be more widespread in Europe due to global change. The elevated risk of reworking historical contamination is one of the yet unrecognized hazards accompanying this change. In the Skalka Reservoir, the drawdown-enhanced inflow erosion could be limited by increasing the water level of reservoir during droughts or by planting species, which could decrease the erosional potential of contaminated sediments.

## 5. Conclusions

Pre-dam topography modeling is crucial for reservoir bottom geomorphological study. The quality of the DSM derived from processed archival aerial imagery is influenced by factors that cannot be omitted in some cases, i.e., the quality and distribution of GCPs influence the accuracy of the DSM and the quality of the raw imagery influences the quality of the DSM. Methods for radiometric and image grain noise removal can be applied, but this does not guarantee smooth and accurate results.

In many cases, old maps represent the only available archival data source describing pre-dam topography. An accuracy test of the (elevation) data derived from the processed map is essential, and the usage of even data considered to be the best available may not meet the quality requirement for accurate pre-dam topography reconstruction.

The extreme drawdowns made it possible to employ topography surveying tools: LiDAR and photogrammetry. The use of LiDAR for mapping of the reservoir inflow is unique, but if it is possible, it brings results of the highest quality. Accuracy analysis of bathymetric data is often questionable in cases of large water reservoirs, but the use of a wide variety of methods allows data cross-validation and accuracy analysis. Geophysical methods such as GPR and profiler can provide results particularly valuable due to imaging of subsurface structures.

The accuracy and reliability of the DEM and DSM in reservoir inflows are vital for the analysis of hazards related to erosion and deposition of young pre-dam and dam reservoir sediments because those sediments are most prone to reworking. It is of environmental concern because delta deposits can contain historical contaminants as actually the studied reservoirs. The erosion of sediments in reservoir inflows is significantly enhanced during drawdowns, which may be caused by either reservoir management or by the seasonal droughts expected to become more common in a changing world. In this study, reliable methods of studying the geomorphic development of reservoirs were successfully tested and demonstrated in two reservoirs, which both shows considerable erosion and deposition dynamics during drawdowns. These methods can be recommended for use elsewhere, as detailed works in inflow sediment dynamics are urgently required in future research.

**Author Contributions:** Jan Pacina processed old maps and archival aerial photographs and derived the pre-dam topography. He prepared (field measurement, flight plans), carried out and processed the photogrammetrical and LiDAR survey. He was involved in bathymetric survey, ground penetration radar (GPR) survey, and sub-bottom profiler survey. His task was processing of all GIS data. Zuzana Lenďáková was in charge of the GPR survey and sub-bottom profiler survey. She processed and interpreted data from these methods. Jiří Štojdl was in charge of the sonar survey including postprocessing of the data, made the sediment sampling and supported the GPR and profiler survey. Tomáš Matys Grygar was in charge of the sediment sampling, his task was the interpretation of gathered and processed data, manuscript mentorship. Martin Dolejš worked on the LiDAR data postprocessing and on the measurement and evaluation of control points used for the LiDAR accuracy analysis. All authors have read and agreed to the published version of the manuscript.

**Funding:** The fieldwork in reservoirs and the airborne surveys were supported by the Czech Science Foundation, project. No. 17-06229S, led by O. Bábek, Department of Geology, Faculty of Science, Palacký University in Olomouc, Czech Republic. The LiDAR survey and the manuscript preparation was supported by the Internal Grant Agency of Jan Evangelista Purkyně University in Ústi nad Labem [grant number UJEP-IGA-TC-2019-44-02-2].

**Conflicts of Interest:** The authors declare no conflict of interest. The funders had no role in the design of the study; in the collection, analyses, or interpretation of data; in the writing of the manuscript, or in the decision to publish the results.

## Appendix A

The datasets used for the analyses are available as: Pacina, Jan; Štojdl, Jiří; Lenďáková, Zuzana; Matys Grygar, Tomáš; Dolejš, Martin (2020), "Data for: Dynamics of sediments in reservoir inflows: a case study of the Skalka and Nechranice reservoirs, Czech Republic", Mendeley Data, v1. http://dx.doi.org/10.17632/9xxrp5kmy5.1.

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
