# Peer review of "Dynamics of Sediments in Reservoir Inflows: A Case Study of the Skalka and Nechranice Reservoirs, Czech Republic"

_ijgi, doi:10.3390/ijgi9040258_

Round 1

Reviewer 1 Report

Paper is very interesting. The text and figures is carefully prepared. I have only minor comments regarding adding some missing citation and discussion related to them.

Line 17.Please add country to the River.

Line 25. Correct “genuine” to “initial” hereinafter.

Line 26. “Lowstand” Did you meant water level lowering (so called drawdown) ? If, so please correct this term hereinafter.

Line 30. Very interesting observation.

Line 31. Please specify the type of contamination.

Line 32-33. Very interesting predictions. I suggests to add few words about potential solutions here and discuss this interesting topic in the discussion. Maybe this negative effects maybe decreased by increasing water level of reservoir during droughts or by planting species which decrease erosional potential of contaminated sediments.

Line 40. This is also very good work on this topic.

https://www.nature.com/articles/s41586-019-1111-9?loc=contentwell&lnk=the-study&dom=section-14

Line 43. Please repleace „evoke” with other word.

Line 47-48. I fully agree.

Line 48. I recommend to change the sentence to “The spatial pattern of sediment deposition is difficult to predict..”

Line 57. I have three recommendations regarding this sentence:

  1. The work which is reference here as [8] is not published yet and even not described in details in the references list.
  2. I think that contamination e.g. by heavy metals is related mainly to finest fractions of sediments which is deposited in the deepest parts of reservoir See e.g., work done below: http://www.pg.geo.uj.edu.pl/documents/3189230/4676035/2006_116_99-110.pdf;Full

https://onlinelibrary.wiley.com/doi/abs/10.1002/%28SICI%291096-9837%28199612%2921%3A12%3C1091%3A%3AAID-ESP653%3E3.0.CO%3B2-2

https://onlinelibrary.wiley.com/doi/10.1111/j.1365-3091.1993.tb01347.x

I recommend to discuss the study setting characteristics (e.g., river sediment grain-size) in the inflow zone with the above works.

Line 90-92. I fully gree with this hypothesis. Evidences from Polish Carpathians Rivers suggests the same. See e.g., https://www.sciencedirect.com/science/article/abs/pii/S0169555X15301537.

Line 105-106. Please give some information from these references here.

Line 107 and 120. Please give information on bed grain-sizes to both study sites.

Line 335. What does it meant “Chyba! Nenalezen zdroj odkazů.

Line 398-401 and 405-406. In discussion please compare this findings with the works which observed lateral channel instability

https://www.sciencedirect.com/science/article/abs/pii/S0169555X15301537 and stability in backwater (https://www.sciencedirect.com/science/article/abs/pii/S0341816217300589) Is your pattern resemble them? In the upper part of the inflow/backwater zones ( so called backwater).

Line 440-444, and line 561-571. Discuss this information with this work. (in discussion section) http://www.pg.geo.uj.edu.pl/documents/3189230/4676035/2006_116_99-110.pdf;Full

568-571. Please discusshere also above mentioned works from Polish Carpathian which describe geomorphic changes related to coarse sediment storage in backwater zone.

584-586. I also recommend to discuss the delta types you were documented with the work of Klimek et al., from 1990. (Klimek K., Łajczak A., Zawilińska L., 1990, Sedimentary environment of the modern Dunajec delta in the artificial Rożnów Lake, Carpathian Mts., Poland, Quaest. Geogr.,11/12.).

  1. Please not use the word „headwater” for delta parts descriptions.
  2. Please add few word about potential action which maybe done to decrease the erosion of contaminated sediments in the future.

Author Response

Thank you for the review. For our responses please see the attachment.

Reviewer 2 Report

The paper topic is appropriate for the journal. Authors assessed the impact of sediments in reservoir inflows by using DEMs from different periods. The issue considered in the manuscript is very important to the geomorphometry community. However, I have several remarks, which should be taken into account before the publication of this article.

To Fig. 1. A higher resolution can be given if possible. There should be markings A where the research area is in the background of the Czech Republic and B where the main map.

To Fig. 2 and 3. If possible, some places can be added because the map is almost blind.

To Fig. 7 A and B. If possible, a higher resolution can be given. Contour layers are heavily torn, it is worth smoothing their appearance

Line 82 - Was Humminbird Helix 7 CHIRP DI GPS G2 sonar equipped with standard GPS or RTK corrections applied?

Line 90 It is worth adding a map showing the location of the profiles - possibly a path from the probe swimming. I also recommend referring to IHO S44 standards for hydrographic surveys in the literature

Line 306 - Some notes in Czech are added

Line 310 - There are several papers in the literature on terrain changes caused by opencast mining. Lapčík, V .; Lapčíková, M 2011, Pandey, A.C .; Kumar, A. 2014 or works on analysis by similar methods at Belchatow coal mine in Poland.

Line 335 - Some notes in Czech are added

Literature No. 8 - needs to be corrected

Author Response

(The authors gave the same response as above.)

Reviewer 3 Report

In the paper the changes in bottom morphology of two reservoirs functioning on one river were investigated. In the analysis a few methods have been used. The authors, despite the fact that they try to show direction and scale of morphological changes of the river valley bottom, focus on the presentation of accuracy of the different methods used in the reconstruction of valley bottom changes over a many decades. Therefore, the paper is very technical. On the one hand, it is very overloaded with technical details showing the pros and cons of methods what make text a little bit difficult for reading and understanding. On the other hand detailed description of methods  is the advantage of this analysis in the context of potential application such research approach in other studies.

In my opinion the paper is scientifically quite interesting and has a good research hypothesis, analysis carried out correctly as well as logical structure of the text. In the conclusions part, the authors should specify more precisely what is their achievement/s.

I see the potential for use of presented scientific approach by other researchers. I fully recommend the reviewed paper for publication.

Author Response

(The authors gave the same response as above.)

Round 2

Reviewer 1 Report

Dear Authors

All my previous concerns were properly addressed in revised version. I think that paper maybe accepted in present form. This is very interesting work. 

All the best,

Reviewer 2 Report

All recommendations have been included in the revised article. Thank you for sending the map with sonar test paths. I have experience in this topic and I know that they are never so equal and book-like. You have written a very good article. Congratulations.